# Effects of Volatile Organic Compounds Produced by *Pseudomonas aurantiaca* ST-TJ4 against *Verticillium dahliae*

**DOI:** 10.3390/jof8070697

**Published:** 2022-06-30

**Authors:** Hang Ni, Wei-Liang Kong, Yu Zhang, Xiao-Qin Wu

**Affiliations:** 1Co-Innovation Center for Sustainable Forestry in Southern China, College of Forestry, Nanjing Forestry University, Nanjing 210037, China; 739703613@njfu.edu.cn (H.N.); k3170100077@njfu.edu.cn (W.-L.K.); qiaozhang@njfu.edu.cn (Y.Z.); 2Jiangsu Key Laboratory for Prevention and Management of Invasive Species, Nanjing Forestry University, Nanjing 210037, China

**Keywords:** *Pseudomonas aurantiaca*, volatile organic compounds, *Verticillium dahliae*, melanin, microsclerotia

## Abstract

*Verticillium dahliae* is one of the most destructive fungal pathogens, causing substantial economic losses in agriculture and forestry. The use of plant growth-promoting rhizobacteria (PGPR) is an effective and environmentally friendly strategy for controlling diseases caused by *V. dahliae*. In this study, 90 mm in diameter Petri plates were used to test the effect of volatile organic compounds (VOCs) produced by different concentrations of *Pseudomonas*
*aurantiaca* ST-TJ4 cells suspension on *V. dahliae* mycelia radial growth and biomass. The mycelial morphology was observed by using scanning electron microscopy. The conidia germination and microsclerotia formation of *V. dahliae* were evaluated. The VOCs with antifungal activity were collected by headspace solid-phase microextraction (SPME), and their components were analyzed by gas chromatography-mass spectrometry (GC-MS). The VOCs produced by strain ST-TJ4 significantly inhibited the growth of mycelium of *V. dahliae*. The morphology of the hyphae was rough and wrinkled when exposed to VOCs. The VOCs of strain ST-TJ4 have a significant inhibitory effect on *V. dahliae* conidia germination and microsclerotia formation. At the same time, the VOCs also reduce the expression of genes related to melanin synthesis in *V. dahliae*. In particular, the expression of the hydrophobin gene (*VDAG*-*02273*) was down-regulated the most, about 67-fold. The VOCs effectively alleviate the severity of cotton root disease. In the volatile profile of strain ST-TJ4, 2-undecanone and 1-nonanol assayed in the range 10–200 µL per plate revealed a significant inhibitory effect on *V. dahliae* mycelial radial growth. These compounds may be useful to devise new control strategies for control of Verticillium wilt disease caused by *V. dahliae*.

## 1. Introduction

*Verticillium dahliae* is a highly damaging soil-borne pathogenic fungus, and it is the causal agent of Verticillium wilt in temperate and subtropical climates. *V. dahliae* has great genetic plasticity and is able to infect more than 200 plant species including cash crops such as cotton, tomatoes, potatoes, and sunflowers [1]. Valuable tree species such as *Canarium* spp., *Sophora*
*japonica,* and *Ulmus pumila* are also included. Plants infected with *V. dahliae* will suffer from wilt, leaf drop, necrosis, V-shaped spots, and yellowing of leaves [2,3]. In severe cases, it can kill plants and cause significant economic losses to agriculture and forestry. Therefore, how to effectively control Verticillium wilt caused by *V. dahliae* has received attention from researchers worldwide [4].

At present, various strategies have been used to prevent the occurrence of Verticillium wilt such as soil amendments, soil fumigation, soil solarisation, tillage, and the development of resistant plant varieties [5,6]. However, *V. dahliae* can survive for a long time in soil as microsclerotia without a host, so these methods are not very effective in controlling *V. dahliae* [1]. In addition, chemical methods such as the use of fungicides can have a negative impact on human health and the environment. Biological control is considered to be the control method with the most potential so far because it reduced environmental pollution, has a wide control range, and lasts for a long time [7,8].

Volatile organic compounds (VOCs) produced by microorganisms are an effective tool in biological control. Microbial VOCs are secondary metabolites with low molecular weight, weak polarity, high vapour pressure, low boiling point, and lipophilic properties [9,10,11]. These characteristics give them the advantages over other secondary metabolites of microbial nature, being able to travel long distances to mediate interactions between organisms that are not in direct contact and to be perceived at low concentrations. Therefore, VOCs are now widely used in the prevention and treatment of plant diseases [12]. Xie et al. [13] found that *Bacillus subtilis* DZSY21 strongly inhibited the growth of *Curvularia lunata* by producing VOCs and that 2-methylbutyric acid, 2-heptanone, and isoamyl acetate in the VOCs of strain DZSY21 were detected to inhibit mycelial growth and conidiophore formation in *C.*
*lunata*. VOCs produced by *Pseudomonas chlororaphis* subsp. *aureofaciens* are effective against black rot of sweet potato tubers caused by *Ceratocystis fimbriata* [14]. Wang et al. [15] found that dimethyl trisulfide and aniline in the volatilome of *Streptomyces alboflavus* TD-1 play an important role in the *Aspergillus flavus* growth inhibition and aflatoxin production. *S*. *alboflavus* and its VOCs have great potential for development into a biopesticide.

*Pseudomonas aurantiaca* ST-TJ4 was isolated and screened from the rhizosphere soil of healthy poplar trees at Tianjin (China) and was found to have a wide range of antifungal effects. The inhibition substances were analyzed and found that the VOCs produced by the ST-TJ4 strain showed a stronger inhibition capacity compared to the diffusible substances [16]. However, whether the VOCs produced by this bacterium have an antagonistic effect on the soil-borne pathogenic fungus *Verticillium dahliae* is not known. The mechanism of action of these antifungal VOCs requires examination. For this reason, this study examined the effect of *P. aurantiaca* ST-TJ4 VOCs on the morphology and physiological activity of *V. dahliae* in vitro. We collected and identified the VOCs produced by *P. aurantiaca* ST-TJ4 and verified them, revealing the antagonistic mechanism of its VOCs against *V. dahliae*, in order to develop new microbial resources to prevent and control disease caused by *V. dahliae*.

## 2. Materials and Methods

### 2.1. Strain and Growth Conditions

*Pseudomonas**aurantiaca* ST-TJ4 (CCTCC, NO: M2020435) [16], was stored in King’s B medium with 50% (*v*/*v*) glycerol at −80 °C for long-term use [17]. *V. dahliae* was previously isolated from the susceptible *Acer truncatum* in Jining, Shandong, China [18]. The fungus was incubated in potato dextrose agar (PDA) medium for 10 days at 25 °C.

### 2.2. Effects of ST-TJ4 VOCs on V. dahliae Radial Growth

The antifungal effect of the VOCs produced by strain ST-TJ4 was evaluated using Petri plates (90 mm in diameter). Different inoculation volumes (10, 30, 60, and 100 μL) were added to one compartment of the I plate, which contained KB agar medium. Strain ST-TJ4 was cultured on KB agar medium at 25 °C for 36–48 h. A suspension (10^7^ cfu mL^−1^) was prepared in sterile KB. *V. dahliae* mycelial discs (7 mm diameter) were scraped off from 10-day-old colonies grown on PDA at 25 °C in the dark and used to inoculate the other compartment of the Petri plate that contained PDA medium. As a control, a compartment with PDA was inoculated with the fungus, while a non-inoculated KB agar medium was used on the other side. All inoculated dishes were sealed with parafilm and incubated at 25 °C for 10 days. When the fungal hyphae of the control reached the edge of the Petri dish, the two vertical diameters of each colony were measured. The inhibition rate was calculated as: (Cd − Td) × 100%/Cd, where Cd is the colony diameter on the control PDA plate and Td is the colony diameter on the treated PDA plate [19]. For each treatment, three replicates were performed, and every experiment was repeated three times.

### 2.3. Effects of ST-TJ4 VOCs on V. dahliae Mycelial Biomass

The overlapping plate assay was used for the experiments aimed at evaluating the effectiveness of ST-TJ4 VOCs in inhibiting *V. dahliae* mycelia biomass production. A 100 μL spore concentration (1 × 10^7^ conidia mL^−1^) was spread on a Petri plate containing a cellophane-covered PDA medium. Fresh prepared (10^7^ cfu mL^−1^) ST-TJ4 bacterial suspension (100 μL) was spread on another plate containing KB medium. The plate with the bacterium was put upside down on top of the plate with the fungus, both without the lid. The plates were sealed with Parafilm-M to avoid the escape of VOCs from the headspace of the bacteria and fungi. Plates were performed in triplicate and incubated in the dark at 25 °C for two days. Plates of *V. dahliae* conidia covered with dishes having KB agar medium with no ST-TJ4 bacterial suspension were used as control. Then, the hyphae on the cellophane were scraped and weighed.

### 2.4. Scanning Electron Microscopy (SEM)

The effect of VOCs released by the strain ST-TJ4 on *V. dahliae* hyphae morphology was observed by SEM (Quanta 200, FEI, Hillsboro, OR, USA), using an overlapping plate of ST-TJ4 cells and *V. dahliae* conidia after 4 days of exposure. The mycelium grown on PDA medium was used as a control. Hyphae were fixed in 4% glutaraldehyde solution for 24 h, then dehydrated with gradient ethanol solutions (30%, 50%, 80%, 90%, and 100%), freeze-dried, coated with gold, and imaged [20].

### 2.5. Effects of ST-TJ4 VOCs on V. dahliae Conidia Germination

*V. dahliae* mycelial discs were inoculated in liquid complete medium (CM) [21], and the culture was shaken at 150 rpm for 48 h at 25 °C. Conidia were collected by filtering through two layers of sterile cheesecloth and diluted with sterile distilled water to a concentration of 5 × 10^6^ conidia mL^−1^. For conidial germination, the method of sealed Petri plate [21] was applied. Conidia suspension (10 μL) was transferred into a Petri dish. Fresh prepared ST-TJ4 suspension (100 μL, 10^7^ cfu mL^−^^1^) was coated on the KB medium and sealed upside down on the top of the Petri dish containing *V. dahliae* spore suspension. Plates of KB medium without ST-TJ4 were used as a control. Each treatment had three replicates, and every experiment was repeated three times. Conidia germination was checked after 12, 24, and 36 h of incubation at 25 °C.

### 2.6. Effects of ST-TJ4 VOCs on V. dahliae Microsclerotia Formation

Aliquots (100 μL) of *V. dahliae* conidia suspension (5 × 10^7^ conidia mL^−1^) were spread on modified oat medium. The method of sealed Petri plate [21] was applied to 100 μL of ST-TJ4 suspension (10^7^ cfu mL^−1^). Plates of KB medium without ST-TJ4 were used as a control. Three replicates per treatment were performed, and the experiment was repeated three times. All plates were incubated at 25 °C, in the dark, for 14 days. The *V. dahliae* microsclerotia formation was checked under a stereo microscope (SteREO Discovery V.20, Oberkochen, Germany).

### 2.7. Effects of ST-TJ4 VOCs on V. dahliae Melanin Genes Expression

*V. dahliae* mycelium was inoculated in CM liquid medium and shaken at 150 rpm, for 48 h, at 25 °C. The suspension was filtered through monolayer cheesecloth and diluted to 10^6^ conidia mL^−1^. Conidia suspension (100 μL, 10^7^ cfu mL^−1^) was spread on the PDA plate covered with sterile cellophane, while a Petri plate of KB agar medium coated with 100 μL of ST-TJ4 suspension (10^7^ cfu mL^−1^) was sealed with Parafilm upside down as a cap. Plates not inoculated with ST-TJ4 VOCs were used as control. After culturing in the dark at 25 °C for 48 h, *V. dahliae* mycelia were collected, and total RNA was extracted with TRIzol reagent according to the manufacturer’s instructions [19]. cDNA samples were prepared using HiScript II Q Select RT Supermix for qPCR (Yisheng, China). Using 1.0 μL cDNA diluted 1:10 as the template. The entire reaction system is conducted on ABI 7500 (Applied Biosystems, Waltham, MA, USA). Five genes (Table 1) related to melanin synthesis of *V. dahliae* were selected. β-tubulin gene was chosen as the reference gene [22]. The 2^−ΔΔCT^ method was used to calculate the relative quantification of gene expression changes [23,24]

### 2.8. Effects of ST-TJ4 VOCs on V. dahliae Melanin Production

The melanin from *V. dahliae* exposed to ST-TJ4 VOCs per 10 days and untreated mycelia were collected, washed three times with distilled water and dried at 85 °C. The intracellular melanin was extracted with 5 mL NaOH and centrifuged at 11,000 rpm for 10 min. The supernatant was transferred to a new clean centrifuge tube, and the OD value at 400 nm was measured [25]. The melanin content (g/L) was calculated as: OD400 × 0.105 × N, where N represents the dilution factor [26].

### 2.9. Effects of ST-TJ4 VOCs on V. dahliae Inoculated Cotton Seedlings

Seeds of the cotton variety “Miaobao 21”, which are sensitive to *V. dahliae*, were sterilized in succession with 75% ethanol for 5 min, 3% NaClO for 10 min, and rinsed eight times with sterile dH_2_O. The sterilized seeds were soaked for 24 h, then dried and transferred to Murashige and Skoog Basal Medium [27] and incubated at low temperature in the dark for two days. Rooted seeds were arranged into a Petri dishes square and then incubated at 25 °C, 60% of relative humidity and a photoperiod of 16 h of light, 8 h of dark under a photon flux of 4000 for 7 days. Plugs of *V. dahliae* were inoculated onto surfaces of cotton roots. For VOCs exposure, a 5 mm in diameter Petri dish containing KB agar medium coated with 100 μL of ST-TJ4 suspension (10^7^ cfu mL^−1^) was inserted into the square dishes sealed with Parafilm. Plates not inoculated with ST-TJ4 VOCs were used as control. The lesion formation was observed over a 72-h. The length of the lesions was measured. Three seedlings were assayed for each treatment. The assays were performed third to determine their reproducibility [28].

### 2.10. Identification of VOCs by GC/MS Analysis

ST-TJ4 strain was inoculated into 250 mL Erlenmeyer flask containing 100 mL of liquid KB medium at a rate of 1% and cultured at 28 °C, 180 rpm for 60 h. To avoid the escape of VOCs, the conical flask was sealed with aluminium foil. Uninoculated KB medium was performed as the control. A 65 μm PDMS/DVB fiber tip was selected for the determination of bacterial VOCs, and the extraction tip was first aged when used for the first time [16]. The aging temperature of the extraction head selected in this experiment was 250 °C, and the time was 30 min. The cultured bacteria sample was shaken and placed in a 40 °C water bath. The needle of the SPME was inserted through the tin foil and extracted for 30 min. The fiber head was inserted into the gasification chamber of the gas chromatograph (Agilent 7000B, Santa Clara, CA, USA). The sample was analyzed at high temperature in the gasification chamber for 3 min.

GC-MS conditions: using Rtx–5 quartz capillary column; He as the carrier gas; 230 °C as the inlet temperature; 40 °C as the initial temperature, keep it for 3 min, increase temperature at 10 °C/min to 95 °C, and then raise temperature at 30 °C/min to 230 °C. Keep 230 °C for 5 min, ion source is EI source; electron energy is 70 eV; spectrum search is searched by Nist 05 and Nist 05 s library [16].

### 2.11. Effects of Synthetic VOCs on V. dahliae Radial Growth

2-undecone, 1-nonanol, 1-undecene, 2-heptanone, and vinyl decanoate standard at 99% purity from Nanjing Zebra Trading Co., Ltd. (Nanjing, China) were selected to evaluate the antifungal activity of ST-TJ4 volatilome in I plates tests. *V. dahliae* was inoculated as a 6 mm mycelial disc on PDA medium on one side of the plate, while 10, 50, 100, or 200 μL of standards were added to the other side. Plates were sealed with parafilm and incubated at 25 °C for 10 days. The antifungal effect was recorded as radial growth inhibition against an untreated control. Each treatment with three replicates was performed, and every experiment was repeated three times.

### 2.12. Statistical Analysis

The data were analyzed by analysis of variance and Duncan’s multiple comparison with SPSS 17.0 software (IBM, Armonk, NY, USA), and the standard errors of all mean values were calculated (*p* < 0.05). Prism 8.0 software (Prism Software, Irvine, CA, USA) was used for drawing.

## 3. Results

### 3.1. Effect of VOCs Produced by P. aurantiaca ST-TJ4 on V. dahliae Growth

VOCs produced by *P. aurantiaca* ST-TJ4 had a significant inhibitory effect on the radial growth of *V. dahliae* colonies (Figure 1A). As the concentration of ST-TJ4 increases, the relative inhibition rate of *V. dahliae* by the VOCs gradually increases (Figure 1B). When *V. dahliae* was co-cultured with 100 μL of ST-TJ4 (1 × 10^7^ cfu mL^−1^) for 8 days, the inhibition rate reached 63.61%.

The fresh weight of *V. dahliae* mycelia exposed to VOCs produced by ST-TJ4 strain was significantly lower than that of the control: 0.10 ± 0.01 and 0.22 ± 0.04 g, respectively. These results showed that the ST-TJ4 strain VOCs can inhibit the mycelial growth of *V. dahliae*.

### 3.2. Effect of VOCs Produced by P. aurantiaca ST-TJ4 on V. dahliae Hyphae Morphology

SEM micrographs (Figure 2) demonstrated that the hyphae of *V. dahliae* without ST-TJ4 treatment were smooth and strong and the conidia were plump, while the surface of the mycelium treated with ST-TJ4 VOCs was rough and wrinkled and the spores were shriveled.

### 3.3. Effect of VOCs Produced by P. aurantiaca ST-TJ4 on V. dahliae Spore Germination

VOCs produced by strain ST-TJ4 inhibit the *V. dahliae* conidia germination (Figure 3A). In each observation time, the germination rate of conidia treated with strain ST-TJ4 volatilome was significantly lower than the untreated control. The lower germination rate (17.87%) was observed after 12 h of exposition (Figure 3B).

### 3.4. Effect of VOCs Produced by P. aurantiaca ST-TJ4 on V. dahliae Microsclerotia Formation

A 14-day exposure to the strain ST-TJ4 VOCs reduced microsclerotia formation by about 8.35-fold (Figure 4).

### 3.5. Effects of ST-TJ4 VOCs on V. dahliae Melanin Formation

Compared with the control (Figure 5A), the expression of melanin formation-related genes after treatment with ST-TJ4 strain VOCs was significantly down-regulated.

Among them, the hydrophobin gene (VDAG-02273) was down-regulated about 67-fold, followed by the 1, 3, 6, 8-THN reductase gene (VDAG-03665) down-regulated about 3.8-fold, while cyclosporine dehydratase (VDAG-03393) and anthocyanin reductase gene (VDAG-00183) were down-regulated 3.62 and 3.04-fold, respectively. The intracellular melanin content of *V. dahliae* after ST-TJ4 VOCs exposition was significantly lower than that of the control (Figure 5B).

### 3.6. Inhibition by the VOCs Produced by ST-TJ4 Strain of Infection by V. dahliae

The VOCs produced by ST-TJ4 strain significantly relieve the severity of root disease on cotton by *V. dahliae* (Figure 6A). The cotton root in the control treated with *V. dahliae* showed brown lesions 72 h after inoculation (Figure 6A). In the presence of VOCs, the cotton root showed only a slight yellow color within a much smaller zone (Figure 6A). The size of the lesions in the VOCs treated cotton root was significantly smaller than the unexposed (Figure 6B).

### 3.7. Identification of VOCs by GC/MS Analysis

GC-MS/MS analysis (Figure 7) differentiates the composition of VOCs between the control and the strain ST-TJ4. Twenty different VOCs were detected in strain ST-TJ4 volatilome (Figure 7, Table 2). The most abundant molecule is 1-undecene (46.26%), followed by 2-undecone (20.82%) and 4-acetylcycloheptanone (13.76%).

### 3.8. Effects of Synthetic VOCs on V. dahliae Radial Growth

Among the 20 volatile compounds produced by the strain ST-TJ4 and assayed against *V. dahliae*, 2-undecanone and 1-nonanol have antifungal activity (Figure 8). In the range 10–200 μL, the effects of 2-undecanone are related to the increasing concentration evaluated. However, 1-nonanol has the complete inhibition at all the tested volumes.

## 4. Discussion

The potential application of volatile organic compounds (VOCs) released by microorganisms as biocontrol attracted attention as an alternative to developing environment-friendly products in controlling plant pathogens. Microbial VOCs are lipophilic and low-molecular-weight carbon-containing compounds, which easily evaporate under atmospheric temperature and pressure conditions. Due to their chemical and physical properties, VOCs diffuse through the atmosphere and soil, and they are considered ideal vectors of “information chemicals” involved in organisms communications [29,30]. Since VOCs are renewable, biodegradable, and have low toxicity compared to chemicals, they could play an effective role in postharvest disease controls [31]. In previous studies, VOCs produced by the *P. aurantiaca* strain ST-TJ4 had strong antifungal effects on plant pathogenic fungi (i.e., *Botryosphaeria berengeriana*, *Colletotrichum tropicale*, *Cytospora chrysosperma*, *Fusarium graminearum*, *Fusarium oxysporum*, *Fusicoccus aesculi*, *Pestalotiopsis versicolor*, *Phomopsis ricinella*, *Rhizoctonia solani,* and *Sphaeropsis sapinea*) and oomycetes (i.e., *Phytophthora cinnamomi*) [16]. VOCs produced by *Pseudomonas fluorescens* ZX affect *Botrytis cinerea* growth and spore germination rates [32].

The biological mechanism of antagonistic bacteria is associated with the characteristics of volatile microbial metabolites. In this study, the diffusible substances produced by *P. aurantiaca* strain ST–TJ4 showed interesting inhibitory effects against radial growth, mycelia fresh weight, conidia germination rate, and microsclerotia formation of *V. dahliae*, the aetiological agent of Verticillium wilt. ST-TJ4 VOCs act their inhibitory effects on *V. dahliae* in a concentration-dependent manner. Our results confirm the inhibitory effects of ST-TJ4 VOCs on plant pathogens [16] and contribute to increased antifungal effects of bacterial VOCs.

In general, water-spread antimicrobial compounds affect the cell surface morphology of target pathogens [33,34]. Volatile microbial substances work in a special fumigation manner.

The effect of strain ST-TJ4 VOCs on *V. dahliae* cell surface was explored by SEM observation. The *V. dahliae* hyphae exposed to ST-TJ4 volatile compounds show wrinkled and twisted morphology. These observed morphological changes could be due to the massive loss of intracellular material [35].

Microsclerotia are dormant structures formed by *V. dahliae* in dying and dead plant tissues after colonization and in response to nutritional cues in a changing environment [36]. It has strong stress resistance and can survive in the soil for more than ten years. Once it encounters a suitable host plant, the microsclerotia can germinate to form hyphae and infect plants. The most significant feature in the formation of microsclerotia is the accumulation of melanin. Melanin improves the stability and the resistance of plant pathogenic fungal cell wall [37,38]. Melanin is related to fungal pathogenicity [39]. In *V. dahliae*, melanin has a protective effect against high temperature stress and UV irradiation [40]. Therefore, the key factor for the prevention the *V. dahliae* is to inhibit the formation of melanin and microsclerotia [39]. Treatments with *Paenibacillus alvei* K165, isolated from tomato root tips, reduced *V. dahliae* microsclerotia germination rate in eggplant root tips and elongation areas [41]. Chaetovirin A, a metabolite produced by *Chaetomium globosum* CEF-082, has a significant effect on the germination of *V. dahliae* microsclerotia [42]. Interestingly, *Bacillus subtilis* lipopeptides inhibit microsclerotia formation in *V. dahliae* [43]. In this study, the VOCs of strain ST-TJ4 inhibited the formation of microsclerotia and melanin of *V.*
*dahliae.*

The synthesis of melanin is a complex process. Five enzymes are involved in the synthesis of melanin from *V. dahliae*. First, acetyl CoA is carboxylated by carboxylase to form malonyl CoA, then polyketide synthase polymerizes malonyl CoA to form 1,3,6,8-THN; 1,3,6,8-THN is catalyzed by 1,3,6,8-THN reductase to form small columnar spirochetes. Cyclosporon is further catalyzed by Cyclosporon dehydratase to form 1,3,8-THN; 1,3,8-THN is catalyzed by anthocyanin reductase to form Cylindroquinone. Cylindroquinone is further dehydrated to form 1,8-THN. Finally, using 1,8-THN as the monomer, catalyzed by laccase, phenol oxidase, peroxidase, and catalase, 1,8-THN is polymerized to form DHN melanin [44]. In this study, the relative expression levels of the hydrophobin gene (*VDAG-02273*) and melanin biosynthesis genes (*VDAG-03665*, *VDAG-03393,* and *VDAG-00183*) after the ST-TJ4 VOCs treatment were all down-regulated. Quantitative analysis of melanin in the treatment and the control gave the same result. It shows that the VOCs of ST-TJ4 can inhibit the accumulation of melanin in *V. dahliae*. At the same time, we found that ST-TJ4 VOCs have a significant inhibitory effect on the formation of microsclerotia of *V. dahliae*.

In order to find out which volatile organic compounds produced by the strain ST-TJ4 have an effect on *V. dahliae*, we used a combination of headspace solid phase microextraction (HS-SPME) and gas chromatography mass spectrometry (GC-MS) to analyze the components of VOCs. In comparison with other sampling techniques, HS-SPME allows the volatile metabolites in the headspace of the bacterial cultures to be preconcentrated prior to analysis by GC-MS [45]. HS-SPME-GC-MS has been used for profiling VOCs from various microbial samples without contaminating the culture or causing damage to living cells [46,47,48]. In our analysis, the main volatile compounds produced by strain ST-TJ4 include 20 kinds of ketones, alkanes, esters, and alcohols. Among them, 2-undecanone and 1-nonanol have antagonistic effects on *V. dahliae*. In previous studies, it has been reported that 2-undecanone has antifungal properties. Báez-Vallejo et al. [49] showed that 2-undecanone is abundant in the volatiles produced by the two bacteria *Bacillus* and *Pseudomonas*, which has a significant antagonistic effect on *Fusarium*. 1-nonanol only accounts for 0.47% of the peak area in the strain ST-TJ4 VOCs. However, relative to the 2-undecanone amounts, only 10 µL of a similar concentration of 1-nonanol inhibited *V. dahliae* at 100%. This indicates some substances are high in content, but their inhibitory effect on pathogenic fungal is not obvious, while some substances are low in content, but their inhibitory effect is very significant [19]. However, whether such substance has a certain impact on human health and what is the best safe dose and concentration is worthy of further study. Furthermore, whether the VOCs produced by *V. dahliae* has an impact on the biocontrol bacteria itself, and how much impact it has on the type and amounts of VOCs produced by biocontrol agent are also worthy of further consideration.

## 5. Conclusions

In conclusion, this study demonstrated for the first time that the VOCs produced by *P.*
*aurantiaca* ST-TJ4 can directly inhibit the spore germination of *V. dahliae*, destroy the hyphae, and inhibit the growth of the hyphae. Additionally, VOCs affect the melanin accumulation of *V. dahliae* and inhibit the formation of microsclerotia. An organic compound with excellent inhibitory effect on *V. dahliae,* which is called 1-nonanol, was found among the ST-TJ4 VOCs. This compound was reported for the first time in the research on the prevention and control of *V. dahliae*, which has extremely important practical significance for the development of new fungicides, so as to better prevent and control Verticillium wilt caused by *V. dahliae*.

## Figures and Tables

**Figure 1 jof-08-00697-f001:**
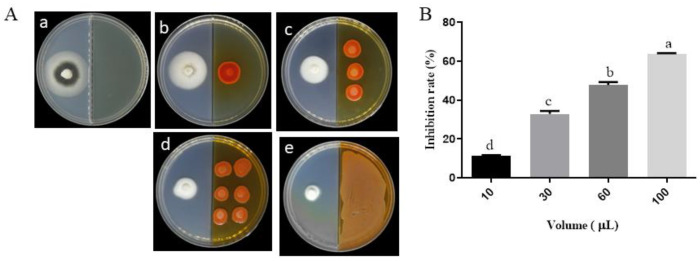
Effects of *Pseudomonas aurantiaca* ST-TJ4 VOCs on *Verticillium dahliae* colonies (**A**) and inhibition rate (**B**) after 10 days of cultures in 90 mm diameter I plates at 25 °C. Pictures in part A are referred to: (**a**) *V. dahliae* colony without ST-TJ4; (**b**–**e**) *V. dahliae* colony treated with 10, 30, 60, and 100 μL of the ST-TJ4 suspension (1 × 10^7^ cfu mL^−1^). In section B, vertical bars represent the standard deviation of the average (*n* = 3). Different lowercase letters indicate significant differences (*p* < 0.05).

**Figure 2 jof-08-00697-f002:**
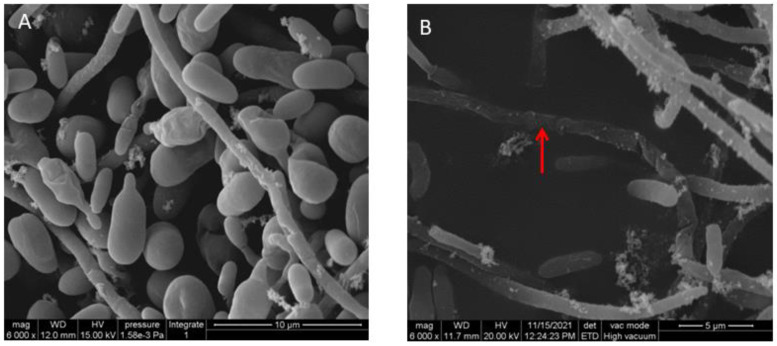
Observations at the scanning electron microscope of *Verticillium dahliae* hyphae in the control PDA plates (**A**) and after 4 days of exposure at volatile organic compounds emitted by *Pseudomonas aurantiaca* ST-TJ4 grown on King B agar medium (**B**). The red arrow indicates the hyphae shrink.

**Figure 3 jof-08-00697-f003:**
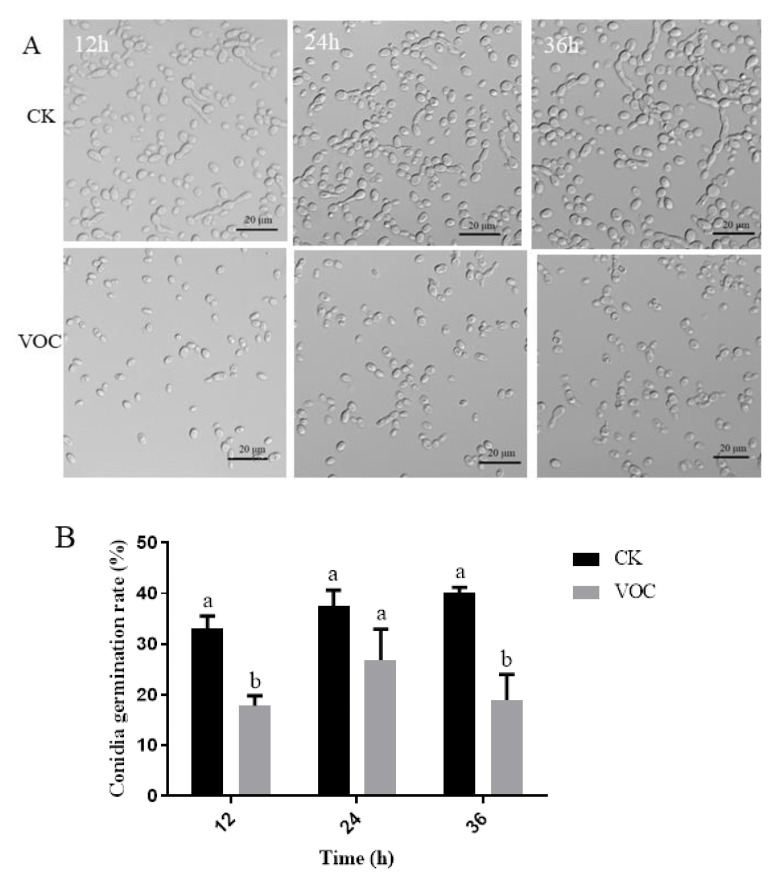
*Verticillium dahliae* conidia germination in overlapping plate assay after 12, 24, and 36 h exposure to liquid KB medium used as control ((**A**), CK) or 100 μL ST-TJ4 liquid cultures in KB medium ((**A**), VOC). Conidia germination rate (**B**). Conidia were considered germinated when the germ tube length exceeded half of the diameter. Vertical bars represent the standard deviation of the average (*n* = 3). Different lowercase letters indicate significant differences (*p* < 0.05).

**Figure 4 jof-08-00697-f004:**
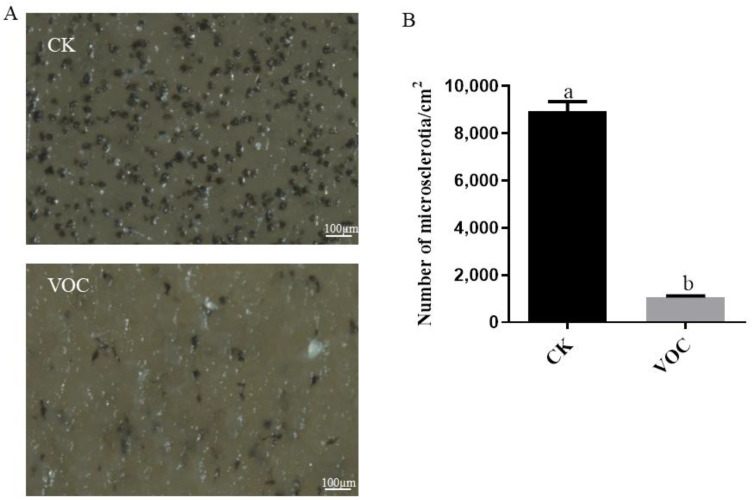
The effect of VOCs of strain ST-TJ4 on *Verticillium dahliae* microsclerotia production: (**A**) Microscopic observation in control (CK) and VOCs treated plate (VOC); (**B**) numerical expression of counted microsclerotia. Vertical bars represent the standard deviation of the average (*n* = 3). Different lowercase letters indicate significant differences (*p* < 0.05).

**Figure 5 jof-08-00697-f005:**
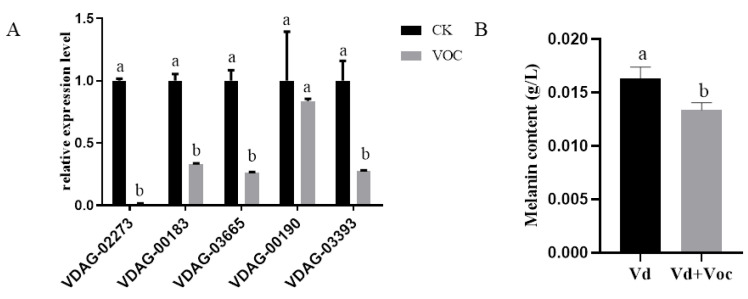
Effects exposure at volatileorganic compounds emitted by *Pseudomonas aurantiaca* ST-TJ4 on *Verticillium dahliae* melanin production: (**A**) gene expression corresponding to Hydrophobin (VDAG-02273), Anthocyanin reductase (VDAG-00183), 1,3,6,8-THN reductase (VDAG-03665), Polymerase (VDAG-00190), and Cylindrosporone dehydratase (VDAG-03393) using RT-qPCR. (**B**) *V. dahliae* melanin concentrations. Each histogram represents the average (*n* = 3). Vertical bars represent the standard deviation. CK = control, VOC = VOCs treated. For each CK/VOC combination, means capped with the same letter are not significant (*p* > 0.05).

**Figure 6 jof-08-00697-f006:**
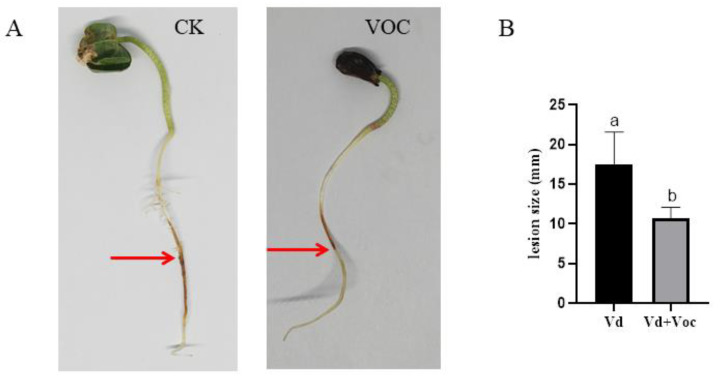
Aspect (**A**) and size (**B**) of lesions on roots of cotton variety “Miaobao 21” inoculated with *Verticillium dahliae* plugs (CK) and exposed to VOCs produced by *Pseudomonas aurantiaca* ST-TJ4. The red arrow indicates the lesion. Vertical bars represent the standard deviation of the average (*n* = 3). Different lowercase letters represent significant differences (*p* < 0.05).

**Figure 7 jof-08-00697-f007:**
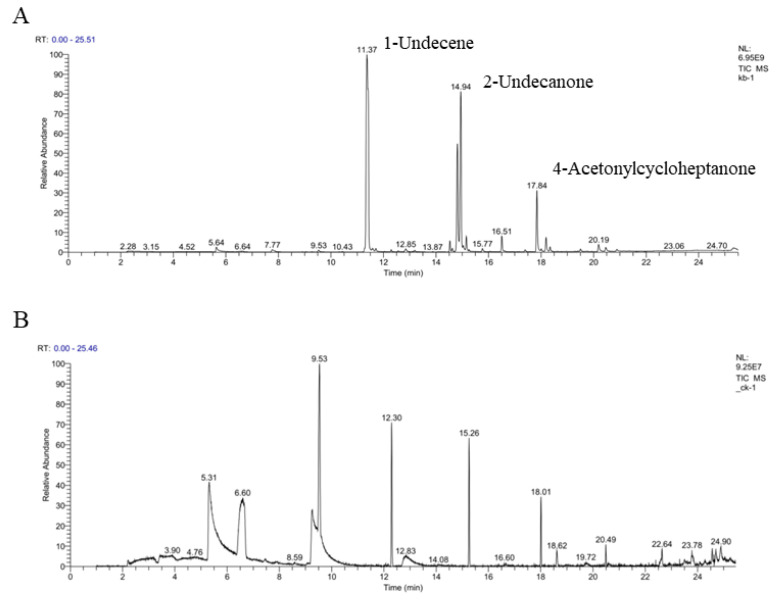
GC-MS analysis of volatile organic compounds produced by *Pseudomonas aurantiaca* ST-TJ4 incubated for 60 h in King B medium (**A**) or uninoculated medium (**B**). Erlenmeyer flasks were incubated in a shaker at 28 °C, 150 rpm.

**Figure 8 jof-08-00697-f008:**
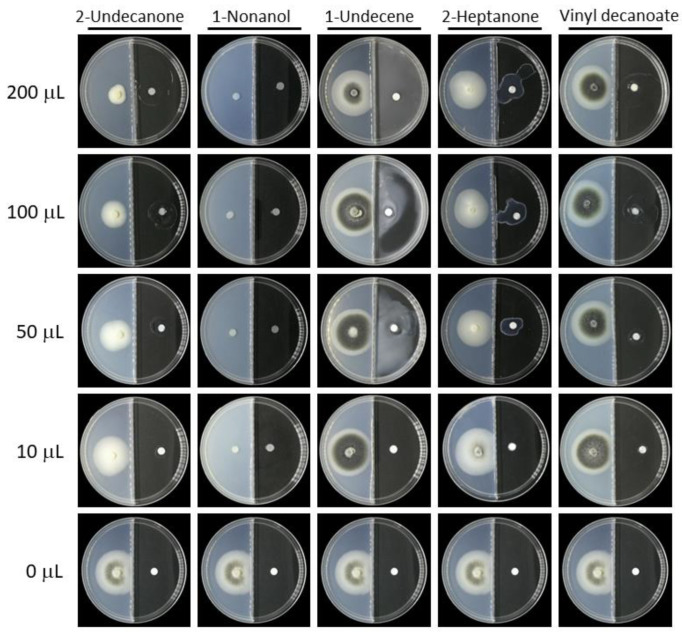
Antifungal activity of 2-undecone, 1-nonanol, 1-undecene, 2-heptanone, and vinyl decanoate associated with volatile organic compounds profile of *Pseudomonas aurantiaca* ST-TJ4 assayed at different concentrations on the growth of *Verticillium dahliae*.

**Table 1 jof-08-00697-t001:** Primers used in RT-qPCR analysis.

Primer	Sequence (5′–3′)	Gene Name
VDAG-00183-F	GAAGGGTGTGACTGTCAATG	*VdT3HR*
VDAG-00183-R	TTGATCCACTCGCAGTCTTC	
VDAG-00190-F	CTTGACGACGTGAACGTTAC	*VdPKS*
VDAG-00190-R	GCTGTCTCAACACAGAGTTC	
VDAG-02273-F	CTATTGCGACGATTGCTCTG	*VdHYD*
VDAG-02273-R	AACGGCCAGACCAAGAATAC	
VDAG-03393-F	GAGTTCCTCGCCATGATCTC	*VdSCD*
VDAG-03393-R	TACTCGCTCCAACGAATCTC	
VDAG-03665-F	AGGTCGTACAAGCCATCAAG	*VdT4HR*
VDAG-03665-R	CTCGCGTGTTGATGTTGAAG	
β-tubulin-F	TTTCCAGATCACCCACTCC	*Reference gene*
β-tubulin-R	ACGACCGAGAAGGTAGCC	

**Table 2 jof-08-00697-t002:** GC-MS/MS volatile profile of strain ST-TJ4.

Retention Time (min)	Relative Peak Area (%)	CAS#	Compound
7.77	0.56 ± 0.02	5874-90-8	2-Heptanone
5.64	0.7 ± 0.04	1066-42-8	Silanediol, dimethyl-
7.77	0.56 ± 0.02	110-43-0	2-Heptanone
11.73	46.26 ± 2.19	821-95-4	1-Undecene
12.85	0.47 ± 0.15	143-08-8	1-Nonanol
14.53	1.1 ± 0.06	NA	5-Isopropenyl-2-methyl-7-oxabicyclo [4.1.0]heptan-2-ol
14.62	0.35	NA	E-1,6-Undecadiene
14.81	13.76 ± 0.65	86428-60-6	4-Acetonylcycloheptanone
14.94	20.82 ± 0.62	112-12-9	2-Undecanone
15.16	1.32 ± 0.12	143-13-5	Acetic acid, nonyl ester
15.77	0.41 ± 0.03	53447-47-5	Lilac aldehyde D
16.51	1.74 ± 0.38	830-13-7	Cyclododecanone
17.84	7.27 ± 0.4	81634-99-3	3-Decen-1-ol, acetate, (Z)-
18.19	1.61 ± 0.15	593-08-8	2-Tridecanone
18.34	0.46 ± 0.03	4704-31-8	Vinyl decanoate
19.5	0.31 ± 0.12	16778-27-1	2(3H)-Benzofuranone, hexahydro-4,4,7a-trimethyl-
20.19	0.98 ± 0.07	56600-19-2	6-Octadecynenitrile
20.46	0.55 ± 0.04	88592-11-4	Hexadecenenitrile
20.89	0.33 ± 0.02	502-72-7	Cyclopentadecanone
25.33	0.81 ± 0.21	630-01-3	Hexacosane

## Data Availability

All datasets generated for this study are included in the article.

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
