# Peer review of "Effects of Volatile Organic Compounds Produced by Pseudomonas aurantiaca ST-TJ4 against Verticillium dahliae"

_jof, 2022, doi:10.3390/jof8070697_

Round 1

Reviewer 1 Report

Dear Authors,

The manuscript jof-1736920-peer-review-v1 “Effects of volatile organic compounds produced by Pseudomonas aurantiaca ST-TJ4 against Verticillium dahliae” presents interesting data on the effects of volatile organic compounds (VOCs) produced by Pseudomonas aurantiaca ST-TJ4 against Verticillium dahliae and their application in new strategies for Verticillium wilt management.

 If I correctly read the paper, the effect of VOCs produced by 10, 30, 60, and 100 μL of ST-TJ4 cells suspensions (1 x 10 7 CFU/mL) of P. aurantiaca strain ST-TJ4 were tested on V. dahliae mycelium radial growth, mat biomass, conidia germination and microsclerotia formations. The effects of VOCs from ST-TJ4 on V. dahliae mycelium morphology were investigated under scanning electron microscopy (SEM) observations. VOCs produced by P. aurantiaca strain ST-TJ4 significantly inhibited the growth of mycelium of V. dahliae in the range of 15-63.61% according to the volumes of tested ST-TJ4 suspension. VOCs have a significant inhibitory effect on V. dahliae conidia germination and microsclerotia formation. SEM observations revealed the presence of rough and wrinkled hyphae when exposed to ST-TJ4 VOCs. Furthermore, the expression of five genes (Anthocyanin reductase, Polymerase, Hydrophobin, Cylindrosporone dehydratase, and 1,3,6,8-THN reductase) related to V. dahliae melanin biosynthesis, was calculated by RT-qPCR analysis using β-tubulin gene as reference. VOCs produced by strain ST-TJ4 reduced the expression of genes related to melanin synthesis in V. dahliae. In particular, the expression of the hydrophobin gene (VDAG-02273) was down-regulated the most, about 67-fold. VOCs can effectively alleviate the severity of root disease on cotton seedlings. Finally, gas chromatography-mass spectrometry (GC-MS) of VOCs, collected by headspace solid-phase microextraction (SPME), revealed the presence of 2-undecanone and 1-nonanol.

My general impression of this study is positive, but its presentation in the form of a manuscript is exposed in a cumbersome way and requires some adjustments and improvements.

I think that the abstract is a very important part of the manuscript, after reading the manuscript I suggest to rewrite it.

 Line 10: What does “tremendous” mean in this sentence?

 The “Introduction” section is pertinent, but the presentation needs to be improved.

Line 35: delete “primary”

Line 45: Is the resistance associated with disease or pathogen?

Line 46: How does V. dahliae survive in the soil without a host?

The “Material and methods” section is confused and difficult to read.

I suggest rewriting.

Verify the correct use of the International System of Units.

Use “Petri” instead of “petri”

Calculate the presence of microsclerotia per cm2 of colony surface.

Insert the reference to “Table 1” in the text.

Insert the acronyms of the genes under the name in table 1 and delete the name of the 5 genes used in the text.

How inoculated cotton seedlings were exposed to VOCs?

Is it possible to set-up an experiment with a combination of the standard compound at concentrations similar to their concentration in VOC?

Which VOCs were selected?

I suggest the following material and method subsections:

            Strain and growth conditions

            Effects of ST-TJ4 VOCs on V. dahliae

                        Radial growth

                        Mycelial Biomass

                        Scanning Electron Microscopy (SEM)

                        Conidia germination

                        Microsclerotia formation

                        Melanin production

                        Melanin genes expression

                        Inoculated Cotton seedlings

            Identification of VOCs by GC/MS analysis

         Effects of Synthetic VOCs Against V. dahliae Radial growth

            Statistical Analysis

Results

This section is also particularly difficult to read. Different concepts are repeated and redundant.

What does "group" mean?

I don’t find data on Mycelial Biomass experiments

Distribute the figures and table 2 appropriately on the page.

Results sections report data from “materials and methods”

If the authors accept the sub-sections proposed in the “Materials and Methods” section, rearrange the division of results.

Figure 1:

Align “A” and “B”

I suggest the legend:

Figure 1. Effect of Pseudomonas aurantiaca ST-TJ4 VOCs on Verticillium dahliae colonies (A) and inhibition rate (B) after ?? days of cultures in 90 mm in diameter I plates at 25 °C. Pictures in part A are referred to: a) V. dahliae colony without ST-TJ4; b-e) V. dahliae colony treated with 10, 30, 60, and 100 μL of the ST-TJ4 suspension (1 x 107 CFU/mL). In section B vertical bars represent the standard deviation of the average (n=3). Different lowercase letters indicate significant differences (p < 0.05).

Figure 2:

I suggest the legend:

Figure 2. Observations at the scanning electron microscope of Verticillium dahliae hyphae in the control PDA plates (A) and after 4 days of exposure at Volatile Organic Compounds emitted by Pseudomonas aurantiaca ST-TJ4 grown on King B agar medium. The red arrow indicates the hyphae shrink.

Figure 3:

Insert the reference time in part A.

Use “Conidia germination rate (%)” instead of “Spore germination rate(%)” in the ordinate axis title.

See the paper “Giorgio, A., De Stradis, A., Lo Cantore, P., and Iacobellis, N. S. (2015). Biocide effects of volatile organic compounds produced by potential biocontrol rhizobacteria on Sclerotinia sclerotiorum. Front.Microbiol. 6:1056. doi: 10.3389/fmicb.2015.01056” as a method reference.

I suggest the legend:

Figure 3. Verticillium dahliae conidia germination in overlapping plate assay after 12, 24 and 36 h of exposure to liquid KB medium used as control (A, CK) or 100μL of fresh ST-TJ4 liquid cultures in KB medium (A; VOC). Conidia germination rate (B). Conidia were considered germinated when the germ tube length exceeded half of the diameter [14]. Vertical bars represent the standard deviation of the average (n=3). Different lowercase letters indicate significant differences (p < 0.05).

Figure 4.

I suggest the legend:

Figure 4. The effect of VOCs of strain ST-TJ4 on Verticillium dahliae microsclerotia production: A) Microscopic observation in control (CK) and VOCs treated plate (VOC); B) numerical expression of counted microsclerotia. Vertical bars represent the standard deviation of the average (n=3). Different lowercase letters indicate significant differences (p < 0.05).

Figure 5:

Improve the magnification of both sections.

Use “Relative intensity” instead of “relative expression level”

I suggest the legend:

Figure 5. Effects exposure at Volatile Organic Compounds (VOCs) emitted by Pseudomonas aurantiaca ST-TJ4 on Verticillium dahliae melanin production: A) mean (± SE; n = 3) gene expression corresponding to Hydrophobin (VDAG-02273), Anthocyanin reductase (VDAG-00183), 1,3,6,8-THN reductase (VDAG-03665), Polymerase (VDAG-00190) and Cylindrosporone dehydratase (VDAG-03393) using RT-qPCR. B) V. dahliae melanin concentrations. Each histogram represents the average (n=3). Vertical bars represent the standard deviation. CK = control, VOC = VOCs treated. For each CK/VOC combination, means capped with the same letter are not significant (p > 0.05).

Figure 6:

Improve the magnification of the “B” section.

I suggest the legend:

Figure 6. Aspect (A) and size (B) of lesions on roots of cotton variety “Miaobao 21” inoculated with Verticillium dahliae plugs (CK) and exposed to VOCs produced by Pseudomonas aurantiaca ST-TJ4. The red arrow indicates the lesion. Vertical bars represent the standard deviation of the average (n=3). Different lowercase letters represent significant differences (p < 0.05).

Figure 7:

Differences in the composition of headspace between the control and the strain ST-TJ4 are not clear.

It seems that the ST-TJ4 modifies the volatile composition of the substrate.

I suggest the legend:

Figure 7. GC-MS analysis of Volatile Organic Compounds in non-inoculated liquid King B medium (B) or after 60 h of Pseudomonas aurantiaca ST-TJ4 growth (A). Erlenmeyer flasks were incubated in a shaker at 28℃, 150 rpm.

Figure 8:

I suggest the legend:

Figure 8. Antifungal activity of 2-undecone, 1-nonanol, 1-undecene, 2-heptanone and vinyl decanoate associated with Volatile Organic Compounds profile of Pseudomonas aurantiaca ST-TJ4 assayed in at different concentrations on the growth of Verticillium dahliae.

The discussion needs some adjustments.

Line 428: use “conclusion” instead of “summary”

Revise the “Reference” section following the Agronomy instructions for Authors.

Reviewer 2 Report

Manuscript ID: jof-1736920 "Effects of volatile organic compounds produced by Pseudomonas aurantiaca ST-TJ4 against Verticillium dahliae" by Ni and collaborators presents a detailed investigation of how VOCs produced by microorganisms can effectively inhibit pathogen growth and protect plant health. The research was well planned and executed, going from a macro level (growth effects in the presence of Pseudomonas aurantiaca ST-TJ4) to a micro level (growth inhibitory effect of 1-nonanol). The research successfully identified promising compounds that can now be further studied and possibly used to protect plants from Verticillium dahliae. Statistical analysis is appropriated and clearly stated in methods and figure legends. The manuscript is well structured but must be reviewed and edited to adjust the text to MDPI standards. As it stands, grammar errors distract the reader and prevent full appreciation of the work. I also have a few minor suggestions listed below to help the authors prepare the revised version. The manuscript will be suitable for publication once the text is improved.

line 38: Delete "which".
line 54: Replace "nature of being" with "nature, being".
lines 119-125: Remove bold formatting.
lines 130-147: Review text and uniform passive voice for clarity.
lines 172-185: Review text and uniform passive voice for clarity.
lines 216-217: It is not clear which compound was used as a reference, please clarify.

Author Response

Response to Reviewer 2 Comments

1: line 38: Delete "which

Response 1: I have deleted it. The revised details can be found in Line 41.

2: line 54: Replace "nature of being" with "nature, being".

Response 2: I have replaced it. The revised details can be found in Line 58.

3: lines 119-125: Remove bold formatting.

Response 3: I have removed it. The revised details can be found in Line 116-122.

4: lines 130-147: Review text and uniform passive voice for clarity.

Response 4: Thank you for your suggestion. I have modified it. The revised details can be found in Line 138-152.

5: lines 172-185: Review text and uniform passive voice for clarity.

Response 5: Thank you for your suggestion. I have modified it. The revised details can be found in Line 191-206.

6: lines 216-217: It is not clear which compound was used as a reference, please clarify.

Response 6: Sorry, I haven’t explained it clear. 2-undecone, 1-nonanol, 1-undecene, 2-heptanone and vinyl decanoate were used to test. I have added them in the text. The revised details can be found in Line 243-244.

Reviewer 3 Report

Studies performed by authors demonstrate that indicated the volatile organic compounds  (VOCs) produced by Pseudomonas aurantiaca ST-TJ4 can directly inhibit the spore germination of V. dahliae, destroying and inhibiting the growth of the hyphae. The volatile organic compounds which appear from the metabolism of  V. dahaliae affect the melanin accumulation in V.dahalie and inhibit the formation of microsclerotia. An organic compound with an inhibitory effect on V. dahliae (i.e.1-nonanol) was found among the VOCs produced by the strain ST-TJ4. This compound has important practical significance for the development of new fungicides, to prevent and control Verticillium wilt caused by V. dahliae.

The manuscript is well written, and the methodology used is correct (but not need so many details), modern techniques are used to analyze volatile organic compounds (GC-MS analysis)  

Before paper publishing, minor modifications are ned to perform in this manuscript, respectively:

1)The Chapter named Materials ad Method must be written more concisely, and more description is needed only in the case in which the authors develop a special or a new methodology. For the rest, if in this chapter the authors used a consecrated methodologies,  here must be added one or more  bibliographic indexes, without so long descriptions; 

2) More attention to typewriting mistakes; the authors must read with the attention their manuscript and made proper corrections.

Author Response

Response to Reviewer 3 Comments

1: The Chapter named Materials ad Method must be written more concisely, and more description is needed only in the case in which the authors develop a special or a new methodology. For the rest, if in this chapter the authors used a consecrated methodologies, here must be added one or more bibliographic indexes, without so long descriptions;

Response 1: Thank you for your precious suggestion. I have made a lot of changes in the original text. The revised details can be found in Line 83-252.

Response 1: Thank you for your suggestion. I have carefully revised each chapter.

2: More attention to typewriting mistakes; the authors must read with the attention their manuscript and made proper corrections.

Response 2: Thank you for your suggestion. I have carefully revised the text.

Round 2

Reviewer 1 Report

Dear Authors,

The rewritten version of the manuscript jof-1736920-peer-review-v2 “Effects of volatile organic compounds produced by Pseudomonas aurantiaca ST-TJ4 against Verticillium dahliae” was improved with respect to the first draft, but some parts are still very confusing.

Suggestions proposed in the previous review were not considered in the revised draft.

The results section usually presents the results of the study, while the comparison with other works is demanded in the discussion section. This manuscript presents a general confusion.

The International System of Units recommend the use of superscript: “cfu mL-1” instead of “cfu/mL”

Uniform “CFU” or “cfu”

Distribute the figures and tables appropriately on the page.

Further suggestions to improve the manuscript.

Line 10: use “substantial economic” instead of “enormous”

Line 13: use “90 mm in diameter I plates” instead of “sterile dichotomous dishes”

Lines 14-15: use “produced by different concentrations of Pseudomonas aurantiaca ST-TJ4 cells suspension on V. dahliae mycelia growth and biomass.” Instead of “produced by … mycelium.”

Line 16: use “conidia” instead of “spores”

Line 17: use “evaluated” instead of “examined”

Line 19: delete “ results showed that the”

Line 20: use “The morphology of the hyphae” instead of “The morphology of the mycelium”

Line 20-21: use “exposed to” instead of “treated with the”

Line 20” delete “It indicated that VOCs can change the morphology of V. dahliae.”

Line 22-23: use “V. dahliae conidia germination and microsclerotia formation.” Instead of “the germination … V. dahliae.”

Line 25: delete “can”

Line 26: use “also detected.” instead “of “found”

Lines 26-27: use “Both molecules assayed in the range 10-200 µL per plate revealed a significant inhibitory effect on V. dahliae mycelial radial growth.” Instead of “were found … V. dahliae.”

Line 44: use “resistant plant varieties” instead of “plant varieties resistant to Verticillium wilt”

As reported in your response, resistance is against the pathogen, then V. dahliae (the fungus) and not Verticillium wilt (the disease).

Line 45 insert “as microsclerotia” between “soil” and “without”

Line 48: delete “has the advantages”

Line 51: use “. Microbial VOCs are secondary metabolites with” instead of “, which have”

Line 56: insert a space between “et al.” and “[”

Line 59: insert a space between “C.” and “lunata”

Lines 60-61: use “black rot of sweet potato tubers caused by Ceratocystis fimbriata” instead of “black rot … potato”

Line 62: use “volatilome” instead of “volatile gas”

Line 63: Use “Aspergillus flavus growth inhibition” instead of “inhibition of Aspergillus flavus growth”

Line 63: use “Streptomyces alboflavus and/or its VOCs” instead of “They”

Lines 80-81: delete “was isolated from the inter-rhizosphere soil of poplar at Tianjin, China”. Lines 65-66 describe this information.

Line 81: use “, “ instead of “. The strain ST-TJ4”

Line 82: delete “The pathogen ”

Line 90: use “. Strain ST-TJ4 was cultured on KB agar medium at 25°C for 36-48 h. A suspension (107 cfu/mL) was prepared in sterile KB” instead of “. .”

Line 88-89: delete “of ST-TJ4 suspension (107 cfu/mL)”

Lines 90-92: use “V. dahliae mycelial discs (7 mm diameter) were scraped-off from ??-days-old colonies grown on PDA at 25°C, in the dark, and used to inoculate the other compartment of I plates which contains PDA medium. As a control, a compartment, with PDA, was inoculated with the fungus, while a non-inoculated KB agar medium was used on the other side. All inoculated dishes” instead of “The pathogen … I plates.”

Lines 94-95: delete “with a vernier caliper”

Line 95: use “was calculated as:” instead of “=”

Line 97: use “For each treatment, three replicates were performed and the ” instead of “Each … every”

Line 100: use “Effects of ST-TJ4 VOCs on V. dahliae Mycelial Biomass” instead of “Mycelial Biomass”

Lines 102-107: use “The overlapping plate assay was used for the experiments aimed at evaluating the effectiveness of ST-TJ4 VOCs in inhibiting V. dahliae mycelia biomass production. A 100 μL spore concentration (1 x 107 conidia mL-1) was spread in a Petri plate containing a cellophane-covered PDA medium. Fresh prepared (107 cfu mL-1) ST-TJ4 bacterial suspension (100 μL) was spread in another plate containing KB medium. The plate with the bacterium was put upside down on top of the plate with the fungus, both without the lid. The plates were sealed with Parafilm-M to avoid the escape of VOCs from the headspace of the bacteria and fungi. Plates were performed in triplicate and incubated in the dark at 25°C for two days. Plates of V. dahliae conidia covered with dishes having KB agar medium with no ST-TJ4 bacterial suspension were used as control. Then, the hyphae on the cellophane were scraped and weighed.”

Lines 110-116: use “The effect of VOCs released by the strain ST-TJ4 on V. dahliae hyphae morphology was observed by SEM (Quanta 200, FEI, United States), using an overlapping plate of ST-TJ4 cells and V. dahliae conidia after 4 days exposure. The mycelium grown on PDA medium was used as a control. Hyphae were fixed in 4% glutaraldehyde solution for 24 h. Then dehydrated with gradient ethanol solutions (30, 50, 80, 90, and 100%), freeze-dried, coated with gold, and imaged [20].” instead of “The effect … [20].”

Line 118: use “Effects of ST-TJ4 VOCs on V. dahliae Conidia Germination” instead of “Conidia Germination”

Lines 119-128: use “V. dahliae mycelial discs were inoculated in liquid complete medium (CM) [21] and the culture was shaken at 150 rpm for 48 h at 25 â—¦C. Conidia were collected by filtering through two layers of sterile cheesecloth and diluted with sterile distilled water to a concentration of 5 x 106 conidia mL-1. For conidial germination, the method of sealed Petri plate [21] was applied. Conidia suspension (??? μL) was transferred into a Petri dish. Fresh prepared ST-TJ4 suspension (100 μL, 107 cfu mL-1) was coated on the KB medium and sealed upside down on the top of the Petri dish containing V. dahliae spore suspension. Plates of KB medium without ST-TJ4 were used as a control. Each treatment had three replicates and every experiment was repeated three times. Conidia germination was checked after 12, 24 and 36 h of incubation at 25℃.” instead of “Mycelial … 25°C.”

Line 130: use “Effects of ST-TJ4 VOCs on V. dahliae Microsclerotia Formation” instead of “Microsclerotia Formation”

Lines 132-137: use “Aliquots (100 μL) of V. dahliae conidia suspension (5×107 conidia/mL) were spread on modified oat medium. The method of sealed Petri plate [21] was applied and 100 μL of ST-TJ4 suspension (107 cfu/mL). Plates of KB medium without ST-TJ4 were used as a control. Three replicates per treatment were performed and the experiment was repeated three times. All plates were incubated at 25°C, in the dark, for 14 days. The V. dahliae microsclerotia formation was checked under a stereo microscope (SteREO Discovery V.20, Oberkochen, Germany).” instead of “We add … Germany).”

Line 139: use “Effects of ST-TJ4 VOCs on V. dahliae Melanin Genes Expression” instead of “Melanin Genes Expression”

Lines 141-154: use “V. dahliae mycelium was inoculated in CM liquid medium and shaken at 150 rpm, for 48 h, at 25℃. The suspension was filtered through monolayer cheesecloth and diluted to 106 conidia/mL. Conidia suspension (100 μL, 107 cfu/mL) was spread on the PDA plate covered with sterile cellophane, while a Petri plate of KB Agar medium coated with 100 μL of ST-TJ4 suspension (107 cfu/mL) was sealed with Parafilm upside down as a cap. Plates not inoculated with ST-TJ4 VOCs were used as control. After culturing in the dark at 25°C for 48 hours, V. dahliae mycelia were collected and total RNA was extracted with TRIzol reagent according to the manufacturer’s instructions [19]. cDNA samples were prepared using HiScript II Q Select RT Supermix for qPCR (Yisheng, China). Using 1.0 μL cDNA diluted 1:10 as the template. The entire reaction system is conducted on ABI 7500 (Applied Biosystems, Waltham, Massachusetts, USA). Five genes (Table 1) related to melanin synthesis of V. dahliae were selected. β-tubulin gene was chosen as the reference gene [22]. The 2−ΔΔCT method was used to calculate the relative quantification of gene expression changes [23,24].” instead of “The mycelium … changes [23,24].”

Table 1.

Align “VDAG-00183-F” with “GAAGGGTGTGACTGTCAATG”

Line 139: use “Effects of ST-TJ4 VOCs on V. dahliae Melanin Production” instead of “Melanin Production”

Lines 158-166: use “The melanin from V. dahliae exposed to ST-TJ4 VOCs per 10 days and untreated mycelia were collected, washed three times with distilled water and dried at 85 °C. The intracellular melanin was extracted with 5 mL NaOH and centrifuged at 11000 rpm for 10 min. The supernatant was transferred to a new clean centrifuge tube, and the OD value at 400 nm was measured [25]. The melanin content (g/L) was calculated as OD400 × 0.105 × N  where N represents the dilution factor [26].” instead of “According … factor).”

Line 168: use “Effects of ST-TJ4 VOCs on V. dahliae Inoculated Cotton Seedlings” instead of “Inoculated Cotton Seedlings”

Lines 169-176: use “Seeds of the cotton variety “Miaobao 21”, which are sensitive to V. dahliae, were sterilized in succession with 75% ethanol for 5 min, 3% NaClO for 10 min and rinsed eight times with sterile dH2O. The sterilized seeds were soaked for 24 hours, then dried and transferred to Murashige and Skoog Basal Medium [27] and incubated at low temperature in the dark for two days. Rooted seeds were arranged into a Petri dishes square and then incubated at 25℃, 60% of relative humidity and a photoperiod of ?? h under a photon flux of ??? for 7 days. Plugs of V. dahliae were inoculated onto surfaces of cotton roots. For VOCs exposure, a 5 mm in diameter Petri dish containing KB Agar medium coated with ?? μL of ST-TJ4 suspension (107 cfu/mL) was inserted into the square dishes sealed with Parafilm. Plates not inoculated with ST-TJ4 VOCs were used as control. The lesion formation was observed over a 72-hour. The length of the lesions was measured. Three seedlings were assayed for each treatment. The assays were performed third to determine their reproducibility [28].” instead of “Seeds … [28].

Lines 180-184: use “ST-TJ4 strain was inoculated into 250 mL Erlenmeyer flask containing 100 mL of liquid KB medium at a rate of 1% and cultured at 28℃, 180 rpm for 60 h. To avoid the escape of VOCs, the conical flask was sealed with aluminium foil. Uninoculated KB medium was performed as the control. A” instead of “The fermentation … experiment, a.”

Line 198: use “Effects of Synthetic VOCs on V. dahliae Radial growth” instead of “Effects of Synthetic VOCs Against V. dahliae Radial growth”

Lines 200-208: use “2-undecone, 1-nonanol, 1-undecene, 2-heptanone and vinyl decanoate standard at 99% purity from Nanjing Zebra Trading Co., Ltd. (Nanjing, China) were selected to evaluate the antifungal activity of ST-TJ4 volatilome in I plates tests. V. dahliae was inoculated as a 6 mm mycelial disc on PDA medium on one side of the plate, while 10, 20, 100 or 200 μL of standards were added to the other side. Plates were sealed with parafilm and incubated at 25°C for 10 days. The antifungal effect was recorded as radial growth inhibition against an untreated control. Each treatment with three replicates was performed and every experiment was repeated three times.” instead of “To test … three times.”

Lines 216-217: use “Effect of VOCs produced by P. aurantiaca ST-TJ4 on V. dahliae growth” instead of “The antagonistic effect of VOCs produced by P. aurantiaca ST-TJ4 on V. dahliae”

Lines 218-223: use “VOCs produced by P. aurantiaca ST-TJ4 had a significant inhibitory effect on the radial growth of V. dahliae colonies (Figure 1A). As the concentration of ST-TJ4 increases, the relative inhibition rate of V. dahliae by the VOCs gradually increases (Figure 1B).” instead of “The mycelial … Figure 1B).”

Lines 224-227: use “The fresh weight of V. dahliae mycelia exposed to VOCs produced by ST-TJ4 strain was significantly lower than that of the control: 0.10±0.01 and 0.22±0.04 g respectively.” instead of “ The wet … V. dahliae” This period starts as a new paragraph.

Line 230: use “Effects” instead of “Effect”

Lines 236-237: use “Effect of VOCs produced by P. aurantiaca ST-TJ4 on V. dahliae hyphae morphology” instead of “Morphological Changes of hyphae After Co-culture of V. dahliae and Strain ST-TJ4”

Lines 238-239: use “SEM micrographs (Figure 2) demonstrated” instead of “To explore … demonstrated”

Line 240: use “conidia” instead of “spore”

Lines: 242-243: delete “The results indicate that ST-TJ4 fumigation can obviously change the morphology of V. dahliae.”

Figure 2: associate Figure and legend on the same page.

Line 250: use “Effect of VOCs produced by P. aurantiaca ST-TJ4 on V. dahliae spore germination” instead of “ST-TJ4 strain VOCs inhibited the spore germination of V. dahliae”

Line 251-254: use “VOCs produced by strain ST-TJ4 inhibit the V. dahliae conidia germination (Figure 3A). In each observation time, the germination rate of conidia treated with strain ST-TJ4 volatilome was significantly lower than the untreated control. The lower germination rate (17.87%) was observed after 12 h of exposition (Figure 3B).” instead of “The results … Figure 3B).”

Line 259: Figure 3 legend: put “Figure 3” in bold.

Line 265: use “Effect of VOCs produced by P. aurantiaca ST-TJ4 on V. dahliae microsclerotia formation” instead of “ST-TJ4 strain VOCs inhibited the formation of microsclerotia of V. dahliae”

Lines 266-270: use “A 14 days of exposure to the strain ST-TJ4VOCs reduce microsclerotia formation of about 8.35 folds (Figure 4).” instead of “The number … microsclerotia.”

Line 279: use “Effects of ST-TJ4 VOCs on V. dahliae Melanin formation” instead of “ST-TJ4 strain VOCs inhibited the formation of melanin from V. dahliae”

Lines 280-281: delete “To better understand the biological effects of strain ST-TJ4 VOCs on melanin production in V. dahliae, we analyzed the differential expression of genes related to melanin synthesis in V. dahliae [29].”

Lines 284-292: use “Among them, the hydrophobin gene (VDAG-02273) was down-regulated about 67-fold, followed by the 1,3,6,8-THN reductase gene (VDAG-03665) down-regulated about 3.8-fold, while cyclosporine dehydratase (VDAG-03393) and anthocyanin reductase gene (VDAG-00183) were down-regulated 3.62 and 3.04-fold, respectively. The intracellular melanin content of V. dahliae after ST-TJ4 VOCs exposition was significantly lower than that of the control (Figure 5B).” instead of “Among … intracellular melanin in V. dahliae.”

Line 305: insert a space between “.” and “The cotton”

Line 306: remove “obvious”

Line 306: use “In” instead of “By comparison, in”

Lines 306 and 308: use Figure” instead of “Fig.”

Lines 308-309: use “The size of the lesions in the VOCs treated cotton root was significantly smaller than the unexposed (Figure 6B).” instead of “Statistical calculations … Figure 6B) [28].”

Line 316: use “Identification of VOCs by GC/MS analysis” instead of “GC-MS/MS Analysis of VOCs Produced by ST-TJ4 Strain”

Lines 317-319: delete “Use a headspace solid phase microextraction (SPME) syringe to collect the VOCs produced by strain ST-TJ4, and analyze it with a GC-MS/MS system. Filter out the same volatiles produced by KB medium.”

Lines 319-324: use “GC-MS/MS analysis (Figure 7) differentiate the composition of VOCs between the control and the strain ST-TJ4. Twenty different VOCs were detected in strain ST-TJ4 volatilome (Figure 7, Table 2). The most abundant molecule is 1-undecene (46.26%), followed by 2-undecone (20.82%) and 4-acetylcycloheptanone (13.76%).” instead of “As shown in (Figure 7 … V. dahliae mycelium.”

Line 329: use “volatile” instead of “VOC”, delete “the”

Figure 7: Align the two chromatographic profiles. Uniform the dimensions

Line 326: Figure 7 legend: put “Figure 7” in bold.

Table 2:

Align “L-Ala-L-Ala-L-AlaVDAG-00183-F” with “2.28”, “0.3±0.04” and “5874-90-8”

Line 331: use “Effects of Synthetic VOCs on V. dahliae Radial growth” instead of “Inhibitory effect of different volumes of standard substances on V. dahliae”

Lines 332-337: use “Among the 20 volatile compounds produced by the strain ST-TJ4 and assayed against V. dahliae, 2-undecanone and 1-nonanol have antifungal activity (Figure 8). In the range 10-200 μL, the effects of 2-undecanone are related to the increasing concentration evaluated. However, 1-nonanol has the complete inhibition at all the tested volumes.” Instead of “The antifungal … dahliae (Figure 8).”

Figure 8:

Verify the dimension of each plate and the position of each picture in the figure. Uniform horizontal and vertical distances among the pictures.

Lines 344-365: use “The potential application of volatile organic compounds (VOCs) released by microorganisms as biocontrol attracted attention as an alternative to developing environment-friendly products in controlling plant pathogens. Microbial VOCs are lipophilic and low-molecular-weight carbon-containing compounds, which easily evaporate under atmospheric temperature and pressure conditions. Due to their chemical and physical properties, VOCs diffuse through the atmosphere and soil, they are considered ideal vectors of "information chemicals" involved in organisms communications [31,32]. Since VOCs are renewable, biodegradable and have low toxicity compared to chemicals, they could play an effective role in postharvest disease controls [33]. In previous studies, VOCs produced by the P. aurantiaca strain ST-TJ4 had strong antifungal effects on plant pathogenic fungi (i.e., Botryosphaeria berengeriana, Colletotrichum tropicale, Cytospora chrysosperma, Fusarium graminearum, Fusarium oxysporum, Fusicoccus aesculi, Pestalotiopsis versicolor, Phomopsis ricinella, Rhizoctonia solani and Sphaeropsis sapinea) and oomycetes (i.e., Phytophthora cinnamomi) [16]. VOCs produced by Pseudomonas fluorescens ZX affect Botrytis cinerea growth and spore germination rates [34].

The biological mechanism of antagonistic bacteria is associated with the characteristics of volatile microbial metabolites. In this study, the diffusible substances produced by P. aurantiaca strain ST–TJ4 showed interesting inhibitory effects against radial growth, mycelia fresh weight, conidia germination rate and microsclerotia formation of V. dahliae, the aetiological agent of Verticillium wilt. ST-TJ4 VOCs act their inhibitory effects on V. dahliae in a concentration-dependent manner.

Our results confirm the inhibitory effects of ST-TJ4 VOCs on plant pathogens [16] and contribute to increase antifungal effects of bacterial VOCs.

In general, water-spread antimicrobial compounds affect the cell surface morphology of target pathogens [35,36]. Volatile microbial substances work in a special fumigation manner.

The effect of strain ST-TJ4 VOCs on V. dahliae cell surface was explored by SEM observation. The V. dahliae hyphae exposed to ST-TJ4 volatile compounds show wrinkled and twisted morphology. These observed morphological changes could be due to the massive loss of intracellular material [37].” instead of “The biocontrol potential … intracellular material [37].”

Lines 371-375: use “Melanin improve the stability and the resistance of plant pathogenic fungal cell wall [39,40]. Melanin is related to fungal pathogenicity [41]. In V. dahliae, melanin has a protective effect against high temperature stress and UV irradiation [29]. instead of “Hensonn et al … UV irradiation [29].”

Lines 377-381: use “Treatments with Paenibacillus alvei K165, isolated from tomato root tips, reduced V. dahliae microsclerotia germination rate in eggplant root tips and elongation areas [42]. Chaetovirin A, metabolite produced by Chaetomium globosum CEF-082 has a significant effect on the germination of V. dahliae microsclerotia [43]. ” instead of “Antonopoulos … V. dahliae microsclerotia.”

Lines 382-383: use “study, ” instead of “experiment, we also found that”

Lines 385-393: verify the punctuation.

Line 393: use “In this study” instead of “In the experiment”

Line 394: delete “[30]”

Line 3954: delete “[29]”

 Align the references 3, 5 and 22.

The references 7, 14, 18, 19, 20, 21, 22, 24, 27, 28, 36 and 44 are not in the Journal of Fungi format.

If it is not necessary, delete DOI information at references 19, 20, 24 and 31.

Reviewer 3 Report

Studies performed by authors demonstrated that indicated the volatile organic compounds (VOCs) produced by P. aurantiaca ST-TJ4 can inhibit the spore germination of V. dahliae, and destroy and inhibit the growth of hyphae. Volatile organic compounds also affect the melanin accumulation of V. dahliae and inhibit the formation of microsclerotia. An organic compound with an excellent inhibitory effect on V. dahliae called 1-nonanol was found among the ST-TJ4 VOCs. This compound,  reported for the first time in the research on the prevention and control of V. dahliae, shows an important practical significance for the development of new fungicides, in order to prevent and control Verticillium wilt caused by V. dahliae.

I read the improved article and I recommend this article for publishing, with minor revision, as follows:

In rows 11; 87 and 89 the words ''I plate'' must be replaced with the words ''Petri plates''.

2) The paper does not have the dedicated Chapter entitled Conclusions. But, the sentences from row 415 can be easily adapted for conclusions. 

Round 3

Reviewer 1 Report

Some minor adjustments:

Pay more attention to the distribution of text between pages

Distribute figures and their legends appropriately on the page.

Line 15: use “mycelia radial growth” instead of “mycelia growth”

Lines 100-101: insert a line

Line 123: insert a beginning of a paragraph after “Germination”

Line 220: insert a beginning of a paragraph after “growth”

Lines 230-231: delete

Line 239: insert a beginning of a paragraph after “morphology”

Figure 2: associate Figure and legend on the same page.

Line 243: delete

Lines 255-256: delete

Lines 268-269: delete

Line 298: delete “.” Between “Figure” and “6A”

Figure 7: Part B is smaller than Part A. Uniform the dimensions

Figure 8: Uniform horizontal and vertical distances among the pictures.

The references 18, 20, 24, 27 and 28 are not in the Journal of Fungi format.